# The Global Textile and Apparel Value Chain: From Mexico–US–China Linkages to a Global Approach

Óscar Rodil-Marzábal [1,*], Ana Laura Gómez Pérez [2] and Hugo Campos-Romero [1]

1   ICEDE Research Group, Departamento de Economía Aplicada, Universidade de Santiago de Compostela, Av. do Burgo, s/n, 15782 Santiago de Compostela, A Coruña, Spain
2   Instituto Politécnico Nacional (IPN), Escuela Superior de Economía (ESE), Av. Plan de Agua Prieta 66, Plutarco Elías Calles, Miguel Hidalgo, 11350 Ciudad de México, CDMX, Mexico
*   Correspondence: oscar.rodil@usc.es

**Abstract:** The aim of this paper is to analyze the participation in the global textile and apparel value chain with special attention, first, to the case of three dynamic and interrelated economies (Mexico, the United States, and China); and second, to a general approach to a larger sample of countries through the analysis of trade in value added. From the descriptive analysis, a high domestic share in each country's exports is found. However, China is the leading exporter in the industry, accounting for around a third of the domestic value added in the global textile final demand. An econometric estimation has also been carried out to observe the effects of tariffs, FDI, and labor costs on the total and backward participation in the textile GVC. In this case, the sample has been extended to 39 developed and 22 developing countries. The results reveal tariff protection as a determinant of the degree of participation of the sector, especially when backward participation and developing economies are considered. However, FDI and labor costs only show the expected results in the case of developing countries. This may be due to the different tasks performed by developing economies (primarily manufacturing) versus developed economies (branding, design) within the sector's value chain.

**Keywords:** global value chain; textile and apparel; international trade; developed and developing countries; forward and backward participation; explanatory factors



## 1. Introduction

The textile and clothing industry is of great importance worldwide. In the case of countries such as Mexico or the United States, it is a relevant activity currently boosted by the advantages of the United States–Mexico–Canada Agreement (USMCA). The significant increase in the demand for textile products has opened the door to complex production processes that met international quality standards and paved the way to produce complete packages. This stimulates the formation of linkages between members of the production chain and provides opportunities for domestic producers. This process has led to the emergence of Global Value Chains (GVCs), a complex phenomenon reflecting the importance of global production linkages for access to new technologies, training, and innovation (Morrison et al. 2008).

The concept of GVCs provides a better understanding of the value creation process and helps understand how this value is captured, held, and leveraged in all industries. The GVC approach offers a global view of the World's industries from two perspectives: governance and upgrading. The former focuses mainly on leading companies and how their supply chains are organized on a global scale. At the same time, the latter involves the strategies that countries, regions, companies, and other actors use to maintain or improve their positions in the global value chain (Gereffi and Lee 2016). From this perspective, the case of the textile and apparel industry can be seen as a clear example of the strategic use of GVCs in a competitive and dynamic business world.

The theoretical definition of GVCs covers the full range of activities required to bring a good or service to the final consumer, from the acquisition of raw materials to delivery to the final consumer (Antrás 2020; Del Prete et al. 2017; Rodil 2017). In this sense, in the context of international fragmentation and the dynamism of production in the textile sector, labor seems to be a crucial factor, especially in manufacturing tasks. However, the progressive cheapening of products means that no country can forever maintain its comparative advantage in producing labor-intensive garments as its economy industrializes and advances (Lu 2018). Furthermore, these GVCs have expanded due to liberalization, the rise of ICTs, and lower transport costs. This has allowed the management of multiple geographically dispersed tasks in a value chain (Baldwin 2016). Thus, the GVC concept covers all value chain stages following an Input–Output structure. It is also defined by a governance structure, which refers to the power relations between the participating firms, and an institutional context, which refers to the local, national, and international political conditions that affect the different stages of the value chain (Gereffi and Fernandez-Stark 2016).

For decades, developing countries have imported parts and components from countries with more advanced technology, although usually only for the assembly of goods sold locally, forming part of a global network (Taglioni and Winkler 2014). However, several developing countries have managed to move up the chain to more advanced and higher value-added tasks (Pahl and Timmer 2020).

Trade in the supply chain is determined by international differences in production and unbundling costs, while technology determines how the different stages of production are linked (Amador and Cabral 2014). For example, a key part of China's success that has allowed it to achieve economies of scale and scope in GVCs is the constant interaction with various nations for the acquisition of inputs and technology to reduce production costs (Gereffi 2019). Thus, GVCs for developing countries are a fast path to industrialization, as internationally fragmented production allows them to join existing supply chains instead of building them, by sophisticating their goods and expanding their product range (Raei et al. 2019).

An essential factor for insertion in GVCs is industrial competitiveness, which is increasingly defined by international production networks (fragmented and spatially dispersed) and less by national borders (Ponte et al. 2019). In this sense, FDI also plays a central role, representing an opportunity for insertion in GVCs for developing countries. However, according to the WTO (2014), not all countries succeed in joining GVCs. Only those whose production is close enough to the global standards of quality and efficiency succeed. Knowledge and technology transfers, usually fostered by FDI and trade openness, tend to trigger the initial integration.

As a key global player, China has shown a trend as the World's leading exporter of manufactured goods and the largest importer of many raw materials, contributing to its status as an important country in the GVCs (Gereffi 2019). Moreover, the increase in Chinese trade in GVCs has been associated with significant changes in wages and employment in China's trading-partner countries (Robertson et al. 2020). Therefore, the dynamics of GVCs depend on the direction of current trade flows (Durand and Milberg 2020). Regardless of the specific type of GVC, the fragmentation of production results in a greater international division of labor and higher specialization gains exploited by the textile industry (Antrás 2020).

Traditionally, the textile sector has been seen as an ideal way for developing countries to enter GVCs. Although markets have become more complex and competitive, the work done by Whitfield et al. (2021) shows that it is still possible to promote industrialization through trade in textiles. This is due to the potential of this activity to generate intra-sectoral networks and generate industrial upgrading trajectories, initially based on a labor cost advantage. Moreover, successful upgrading processes can lead to greater resilience of companies to external shocks (Choksy et al. 2022).

The process of value creation in different countries generates a comparative advantage and a new division of labor, produces new sources for the flow of trade, and increases the level of innovation during the production process, where the main sources of value added are the industries. Therefore, according to Rodil (2017), measuring trade in value added

is a fundamental tool for analyzing international trade in this fragmented context. This methodology is based on the decomposition of gross trade into value-added flows that capture the way and intensity in which international productive fragmentation affects the participating countries. Likewise for Banga (2014), domestic and foreign value added is created during manufacturing, so value-added exports will differ from gross exports and can be estimated by subtracting foreign value added.

Value-added trade is a series of measures that provide a better understanding of production networks and supply chains through statistical data. Thus, for this measure of trade, several indicators assess the participation of countries within the GVC: the backward participation index, which indicates the share of foreign value added as a percentage of gross exports; the forward participation index, which indicates the share of domestic value added embodied in foreign exports as a percentage of gross exports; and the total participation index, which is the sum of former.

This paper aims to analyze the participation of countries in textile and apparel GVC with special attention, first, to the case of three dynamic and interrelated economies (Mexico, the United States, and China); and second, extending the analysis to a larger sample of countries in the textile and apparel GVC through trade in value-added approach. The first part focuses on the changing role of the three selected economies, on their performance as value-added suppliers of the final global demand for textile products, and, especially, on verifying the rise of Chinese leadership in this global industry. Meanwhile, the second part includes an econometric analysis with panel data (61 countries, 24 years: 1995–2018) of some relevant factors explaining this GVC participation.

The main source of data is the TiVA database (December 2021 edition) provided by the Organization for Economic Co-operation and Development (OECD), which provides information on trade in value-added for 66 economies and 45 industrial sectors, covering the period of 1995–2018. Such information can be used, among others, to analyze the integration of economies into GVCs, as well as the country of origin of the value-added embodied in gross trade flows and final demand. Other databases used are UNCTAD for data on FDI flows, and the WTO for data on textile tariff rates.

The remainder of the paper is structured as follows. Section 2 describes the paper's methodology, highlighting the usefulness of trade in value-added approach for analyzing country participation in GVCs. Section 3 presents and discusses the empirical results, analyzing the participation of Mexico, the United States, and China in GVC from a general (all industries) and sectoral (textiles and apparel) perspective. It also analyzes the contribution of these countries as value-added suppliers to the World's final demand for textile products and adopts an extended econometric analysis with panel data (61 countries, 24 years: 1995–2018) to explore relevant factors explaining the participation of countries in this GVC. Finally, Section 4 presents the conclusions of the paper.

## 2. Data and Methodology

The empirical study of GVC participation has a growing number of works analyzing the role played by various explanatory factors (among others, Rahman and Zhao 2013; Arrighetti et al. 2014; Stehrer and Stöllinger 2015; Kowalski et al. 2015; Jona-Lasinio et al. 2016; Vrh 2018). However, the analysis of GVCs from a macroeconomic perspective usually follows the work of Koopman et al. (2014). Their methodology decomposes a country's gross exports into nine components of trade, providing several indicators. These include forward (export-linked) and backward (import-linked) participation indices, the sum of which is considered an indicator of total GVC participation (see Appendix A for the corresponding OECD TiVA indicators). This methodology allows for the tracing of each country's value-added flows to its final consumption destination.

The local supply of intermediate products is one of the main direct export channels attracting FDI, and specialization in the early stages is associated with the production of local inputs obtained by foreign investors (Amendolagine et al. 2017). Hence, one aspect to be considered as a possible explanatory factor for participation in GVCs is the degree

of tariff protection, as this factor acts as a barrier to trade flows, among which trade in intermediate products associated with the GVC linkages is becoming increasingly essential. Thus, it is interesting to verify if there is a negative relationship between the level of tariff protection and participation in GVC.

As Yi (2003) points out, vertical specialization may have enhanced the reduction in tariff rates. Through this strategy, characteristic of GVCs, countries specialize in certain stages of a product's value chain. As a result, a slight reduction in tariff rates has multiple multiplier effects on trade growth. Conversely, increasing tariff rates can reduce trade in GVCs as parts and components pass multiple times across different national borders (OECD 2013).

Among the explanatory factors of GVC participation, FDI stands out as a determining element when analyzing the insertion of countries in the framework of international productive fragmentation. In this regard, various studies (Stehrer and Stöllinger 2015; Kowalski et al. 2015) point to a positive relationship between inward FDI stock and participation in GVC. However, no conclusive results can be found in the literature on the role played by outward FDI stock. Therefore, studying the relationship between FDI and GVC participation is interesting. In general, it is assumed that there is a positive relationship between them. This hypothesis is based on the role of multinational companies as major actors in GVCs.

Another explanatory factor of GVC participation is the labor cost, since labor has traditionally been a critical factor, especially in manufacturing or assembly tasks, usually offshored to developing countries. However, the progressive cheapening of global products has led to an unstable competitive framework (Lu 2018), and the explanatory relevance of this factor may sometimes be unclear. Hence, it is also interesting to analyze the influence of labor costs on countries' participation in GVCs.

Based on these assumptions, an econometric model has been estimated using panel data. This empirical analysis considers a group of 61 countries at different development levels, observed for 24 years (1995–2018). The division of the 61 countries into two development groups is based on the World Bank's most recent criteria (2021–2022). Countries classified as "high income" have been considered developed countries. All other cases have been included in the group of developing countries. This division divides the sample into two groups of 39 and 22 countries, respectively (see Appendix B). The general model to be estimated is specified as follows:

$$\gamma_{it} = \beta_0 + \beta_1 TARIF_{it} + \beta_2 FDI_{it} + \beta_3 LABC_{it} + \varepsilon_{it} \tag{1}$$

where *i* refers to the country and *t* refers to the period. Two dependent variables have been considered for estimation: total participation in GVCs (TPART), expressed as a percentage of gross exports, and backward participation (BPART), also expressed as a percentage of gross exports.

A total of four independent variables have been selected. The two first regressors are TARIF1 and TARIF2, which refer to the average tariff on textile raw materials and the main textile products, respectively. TARIF1 refers to 51 and 52 textile raw material groups and TARIF2 refers to 61 and 62 textile product groups, according to HS classification. Due to multicollinearity problems between both variables, two different models are considered: Model I, including only TARIF1 as the tariff variable, and Model II, including only TARIF2.

The other two independent variables considered are FDI, which refers to the inward foreign direct investment stock, expressed as a percentage of GDP, and LABC, which is the labor cost, expressed as a percentage of value-added. Except for FDI, all variables refer to the textile sector (ISIC Rev.4 codes 13, 14, and 15). FDI is obtained from UNCTAD, labor cost and GVC share variables are obtained from TiVA (OECD 2021), and TARIF data is obtained from WTO.

Therefore, the two considered models are as follows, where the expression relating to the dependent variable (PART) is a generic expression of the GVC participation, which can refer indistinctly to total participation (TPART) or backward participation (BPART):

$$\text{Model I}:\ PART_{it} = \beta_0 + \beta_1 TARIF1_{it} + \beta_2 FDI_{it} + \beta_3 LABC_{it} + \varepsilon_{it}$$
$$\text{Model II}:\ PART_{it} = \beta_0 + \beta_1 TARIF2_{it} + \beta_2 FDI_{it} + \beta_3 LABC_{it} + \varepsilon_{it}$$

(2)

The consideration of backward participation in GVCs (BPART) is due to its relevance for most developing economies, which, in the textile sector, tend to take on manufacturing tasks of lower value added, relative to other tasks, such as garment design and conception.

The reason for using panel data is motivated by the suspicion that participation in GVC is influenced by unobservable factors that correlate with observed variables, such as the factors mentioned above. Therefore, it is assumed that the panel techniques contribute to obtaining consistent estimates of the effect of the variables observed, offering greater possibilities at the time of facing the usual problems in this type of empirical approach.

The joint significance of differing group means and Breusch-Pagan statistic tests point to a panel data structure. The Hausman statistic test points to a fixed effects model. One of the immediate implications of this is that the error term $\varepsilon_{it}$, in Equations (1) and (2), is now broken down into two different effects: a specific country effect ($m_i$) and the remaining error ($v_{it}$). A relevant advantage of this econometric technique is that it allows us to obtain unbiased estimators.

## 3. Results

### 3.1. The Participation of Mexico, the United States, and China in GVC: General Perspective

International trade allows economies to integrate and increase their participation in GVC trade flows, so that activities along a value chain can be carried out by FDI or outsourcing (Kowalski et al. 2015). For example, in apparel, China has been the most dynamic exporter worldwide in clothing since 2001, when it joined the World Trade Organization (WTO).

It adopted a position that has not favored Mexico, due to the displacement of U.S. imports from that country with such a growth that they surpassed Mexican imports (Pino 2020). As a result, in 2003, Mexico dropped from first to second place as a textile and clothing supplier because of its dependence on the economic ups and downs of the United States (Rodríguez and Fernández 2006).

Trade through GVCs offers opportunities to developing countries, especially smaller ones, for global integration, changing the nature of competitiveness (Pathikonda and Farole 2017). This is because much of the labor-intensive production moved to the World developing in the last wave of globalization, with textiles being highly tradable products (Lund et al. 2019).

As can be seen in this paper, the analysis of the participation of Mexico, the United States, and China in the global textile and apparel value chain presents relevant changes in the last decades. Lu (2013) points out that one of the reasons for these changes is that a country's apparel industry gradually upgrades following the path of Cut, Make and Trim (CMT), Original Equipment Manufacturing (OEM), Original Design Manufacturing (ODM), or Original Brand Manufacturing (OBM). In the case of Mexico, for example, the textile industry has been transforming by assuming mainly assembly tasks (e.g., cutting and sewing) and abandoning a series of risk- and knowledge-intensive coordination and design tasks (Pipkin and Fuentes 2017).

The comparative analysis of the participation of Mexico, the United States, and China in the GVC in the period of 1995–2018 shows different participation levels, insertion patterns, and trends. In this sense, Figure 1 provides a clear picture of the different patterns observed. The first difference refers to the total participation rate, with high participation in Mexico in 2018 (46.4% of gross exports), compared to China (36.6%) and the United States (35.6%) but at the same time with a strong predominance of backward linkages in Mexico (35.9% of gross exports), compared to China (17.2%) and the United States (9.5%).

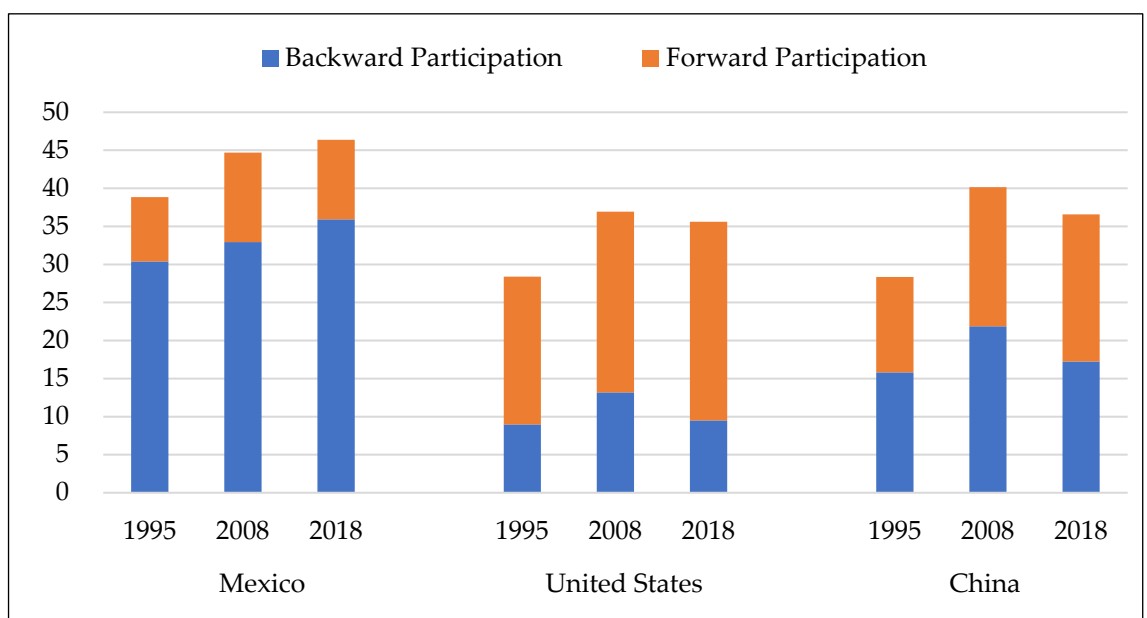

**Figure 1.** GVC participation index (%). Total participation (all sectors) 1995–2018. Source: Authors based on TiVA (OECD 2021).

However, this gap in the level of total participation differs significantly from that observed more than a decade earlier (2008), when Mexico started from a higher level (44.7%) than China (40.2%) and the United States (36.9%). Furthermore, another difference is given by the opposite trends observed in GVC participation; that is, Mexico's participation increased by more than seven percentage points during the study period, while the U.S. and China decreased their participation in GVC between 2008 and 2018 (even though their participation rates are higher in 2018 than in 1995).

The observation of the predominant type of production linkage is fundamental since this analysis is given by decomposing the total participation in its two components: backward and forward participation. Thus, the predominance of China's forward participation (12.6% of gross exports) in 1995 increased by more than six percentage points by 2018 (19.3%); in comparison, Mexico increases by two percentage points from 1995 to 2018 (from 8.5% to 10.5%) while the U.S. shows an increase of more than six percentage points (from 19.4% to 26.1%).

China's rapid growth has made it a major player in virtually all goods produced in GVCs, accounting for 20% of global gross output (Lund et al. 2019), which was initially due to cheap Chinese labor due to low wages (Gereffi and Memedovic 2003). In this sense, one of the causal factors contributing to the reduction in costs and the increase in production rates has been the supply of cheap Chinese labor, which brings low wages (Gereffi and Memedovic 2003).

However, the trend observed for Mexico reveals that its foreign trade operates more as a carrier of value added originating in other countries than as a channeler of domestic value added to later stages of production in the framework of international fragmentation of production. This is, to some extent, a direct consequence of China's productive strategy of gradually substituting foreign value added for domestic value added (Rodil 2017).

Therefore, the reduction in the intensity of participation in GVCs is due to the deepening of the domestic division of labor and the lengthening of domestic value chains (Li et al. 2019). In this sense, the GVC participation trends in Mexico, the United States, and China offer an interesting perspective on their behavior in the GVCs of developed and developing countries, highlighting the case of Mexico's backward linkages that are characteristic of a manufacturing country.

This predominance of backward participation could be associated with countries' participation in production stages close to the final demand, for example, in the case of

assembly tasks (assembly line). However, it is also important to note that countries can show high rates of backward participation by doing non-manufacturing activities that generate high value-added, related to marketing and distribution (Rodil and Gómez 2021).

### 3.2. Participation of Mexico, the United States, and China in GVC of Textiles and Apparel: Sectoral Perspective

### 3.2.1. GVC Participation in the Textile and Apparel Industry

The analysis of the GVC participation at a sectoral level, focusing on the textile and apparel industry (Figure 2), provides a different picture from the general perspective (Figure 1). Firstly, the level of participation is higher when textile GVC is considered. Secondly, the trends at the sectoral level change for Mexico and China, with decreasing and increasing levels of participation, respectively, from 2008 to 2018. Thirdly, the sectoral approach shows a clearer pattern of participation by country, with a deeper forward GVC participation for China, the United States, and Mexico confirming the predominance of backward insertion.

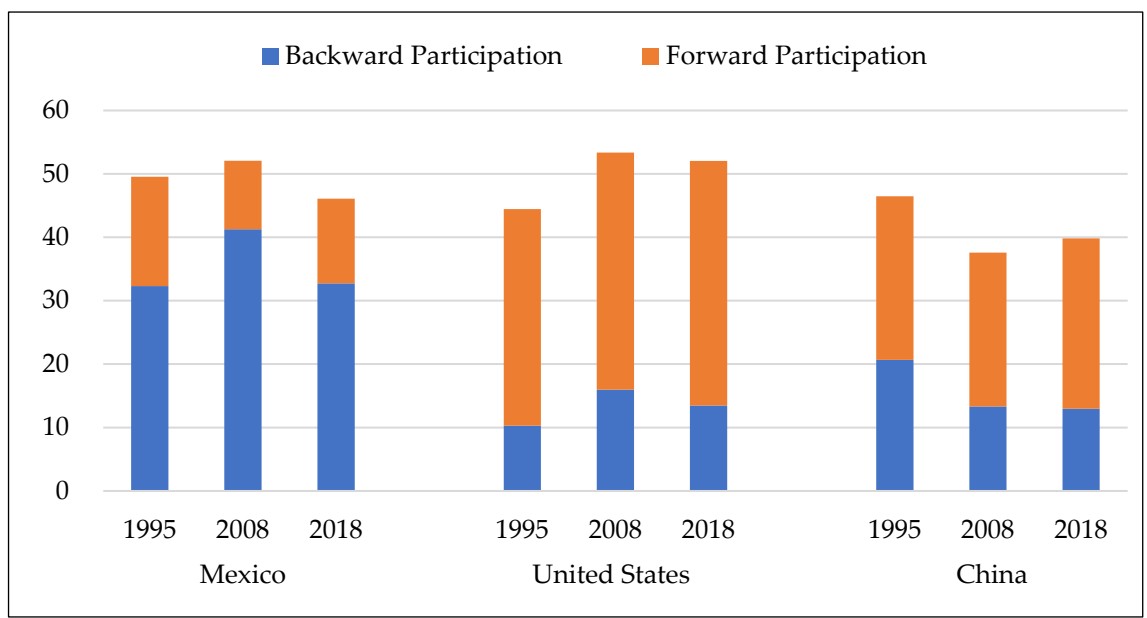

**Figure 2.** GVC participation index (%). Sectoral participation (textiles and apparel) 1995–2018. Source: Authors based on TiVA (OECD 2021).

One of the reasons for the dispute over the North American market between Mexico and China is due to the competition generated by U.S. manufacturing exports. There are two reasons for this rivalry. On the one hand, Mexico is strategically close to the U.S., geographically. On the other hand, China has the advantage of scale, being the world's largest exporter of manufactured goods, especially consumer goods such as textiles and electronics (Gereffi and Luo 2015).

In 2008, China's share of world textile exports was already 26.1%, while the United States and Mexico accounted for only 5% and 0.8%, respectively (Gracia-Hernández 2011). With a national strategy of considering the Fiber–Textile–Clothing Chain (CFTV) as one of the 12 priority branches of the textile industry, Mexico increased its export levels in cotton textile fibers from 7.2% in 2001 to 21.8% in 2010, showing its competitive potential (Vázquez et al. 2015).

Moreover, Mexico's privileged position of sharing more than 3000 km of border with the United States and the existence of free trade agreements account for its progressive trade liberalization, which results in the existence of cheap labor, more than that of all countries, except Asia (Montón 2015).

These aspects provide, in a way, a guarantee for Mexico to position itself in the U.S. market as a continuous development of its insertion in the more sophisticated GVCs. However, its direct participation in the fiber (yarn) and apparel (garment) links has unleashed a continuous cause of tension between China and Mexico (Chen and Goodman 2018). In this sense, some authors (Chen and Goodman 2018) propose that China and Mexico should develop a strategic partnership focused on cooperation by actively seeking business opportunities between them and being more understanding rather than showing their competitiveness with each other.

Therefore, it is interesting to analyze the textile sector's participation in the GVC through the study of value-added trade. The purpose of this analysis is to verify the main research question underlying this paper: is China consolidating its leadership as the leading supplier of value added to the global textile and apparel value chain developing an increasingly important role as a value creator in this global industry? To address this question, changes affecting the origin of value added embodied in textile exports from Mexico, the United States, and China are analyzed below.

3.2.2. Origin of the Value Added of Mexico's Textile and Apparel Exports

Although the origin of the value added embodied in Mexico's textile exports is mostly domestic, it is worth noting that 33% of the total value added exported in 2018 came from abroad, with the United States (14.4%) and China (6.6%) standing out in this share of value added (Figure 3).

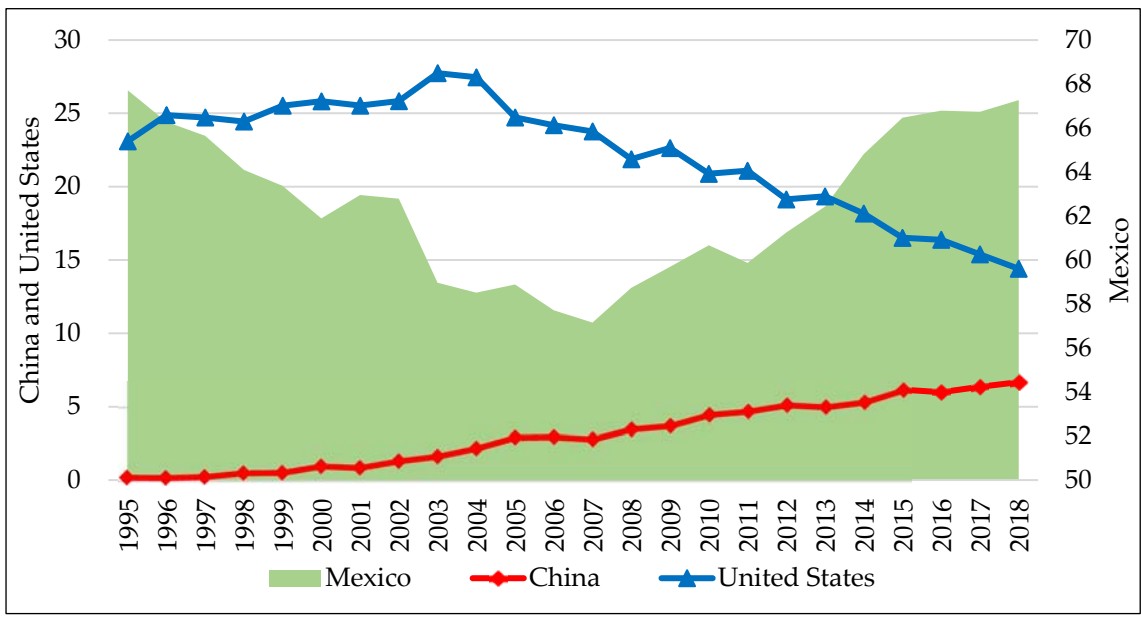

**Figure 3.** Origin of value added in Mexico's textile and apparel gross exports (%). 1995–2018. Source: Authors based on TiVA (OECD 2021).

In any case, what is most striking are the contrasting trends observed in the participation of the United States and China during the 1995–2018 period. In this regard, the U.S. share decreased by more than eight percentage points, while China's share increased by more than 6 percentage points during the same period, more than doubling its initial share. This increase in added value could imply a faster upgrading of activities performed in GVCs and the deepening of intra-product specialization brought about by the recovery of cross-country, production-sharing activities (Li et al. 2019).

Likewise, this increase could also be due to the increased identification of GVC conditions ranging from sourcing cheap labor inputs and basic assembly activities with cheap and unskilled labor, to more advanced forms of value production, such as the full package strategy (Fernández and Gereffi 2019).

The importance of the U.S., in terms of value-added incorporated in Mexican textile exports, stems from the territorial proximity and the productive historical interconnection forged between the two economies over time. A significant proportion of the manufactured products exported by Mexico are made up of dynamic products for world trade, such as textiles and clothing (Fuji et al. 2005).

The implication of the Mexican textile industry in GVCs has been attributed to high transportation costs that fragmented the domestic market and generated a geographically dispersed industry (Gómez-Galvarriato 1999). Likewise, the growth of garment production in China affected world markets by offering the possibility of a greater quantity of textile products being produced through a fragmented process (Robertson et al. 2020).

However, the textile industry in Mexico has obeyed the geographical proximity of the United States, which translates into lower transportation costs, as well as the ease of supplying foreign plants with machinery, components, and materials in general, in addition to the fact that specialized labor represents greater agility when required in Mexico (Hansen 2020).

Therefore, some of the factors that potentially explain the increase in China's share as a source of value-added embodied in Mexican textile exports are: (1) China's lower labor costs compared to Mexico; (2) China's exploitation of economies of scale through investments in infrastructure and transportation logistics that accelerate the commercialization of its exported products (Gereffi and Luo 2015); (3) China's coherent and multidimensional scaling strategy for diversifying its industrial composition which adds high value-added activities (Frederick and Gereffi 2011); and (4) the use of FDI to promote continuous learning in industries as well as leveraging domestic market knowledge.

### 3.2.3. Origin of the Value Added of the United States Textile and Apparel Exports

U.S. textile and apparel exports show a somewhat irregular variation from 1995 to 2018, decreasing almost three percentage points (from 89.7% to 86.5%). The participation of foreign companies, the introduction of new technologies, the continuous training of employees, and the growing competitiveness and innovation are causes that did not foresee an ideal integration of the textile-garment production (or supply) chain. This lets the full package strategy be carried out through selective alliances with leading Mexican companies.

Therefore, the small- and medium-sized companies that make up most of the Mexico sector have been affected. As a result, Mexico's participation in international trade has decreased (Arroyo and Cárcamo 2010). For example, Figure 4 shows a half percentage point share of Mexico in 1995 in the value added embodied in U.S. textile and apparel exports, which barely increased to 0.76% in 2018.

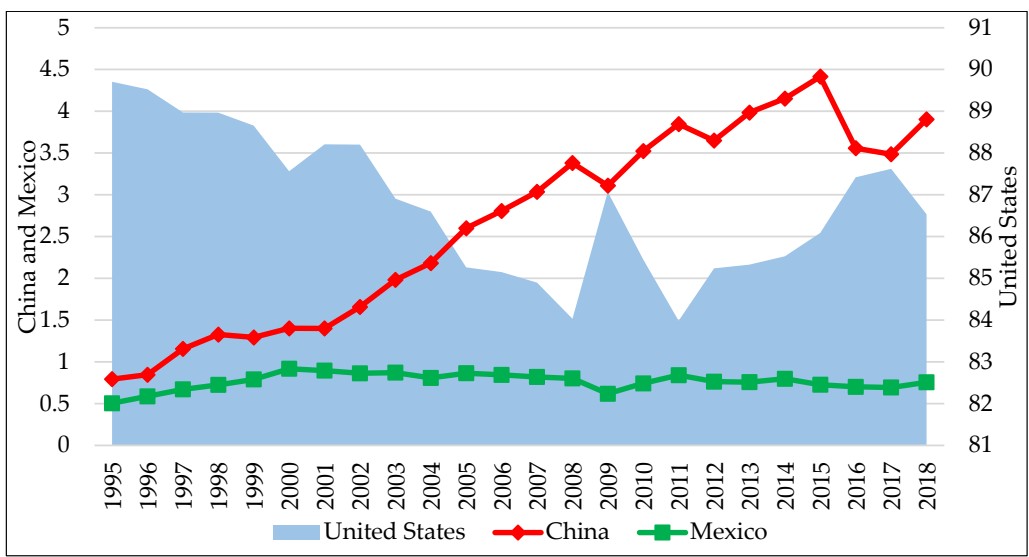

**Figure 4.** Origin of value added in the United States' textile and apparel gross exports (%). 1995–2018. Source: Authors based on TiVA (OECD 2021).

As for China's share, it shows an increase of more than three percentage points from 1995 to 2018 (0.79% to 3.90%), generating a considerable increase in the level of Chinese world exports through its share of the World export market in less than four decades (Gómez Chiñas and García 2017).

Moreover, China's upward trend contrasts both with the fall of domestic (U.S.) value added in the period of 1995–2018, as well as Mexico. This trend has caused the relative importance of China as the source of value-added embodied in U.S. textile exports to increase almost fivefold in 1995.

### 3.2.4. Origin of the Value Added of China's Textile and Apparel Exports

The origin of the added value of China's textile exports is due to the efficiency of the full package strategy in Asia, as it creates competitive advantages that highlight the manufacturing of textile products. This generates around 50% of the final product costs which makes Mexico's participation less than one percentage point and almost constant from 1995 to 2018, as seen in Figure 5 (Castro-Gonzáles and Mathews 2013).

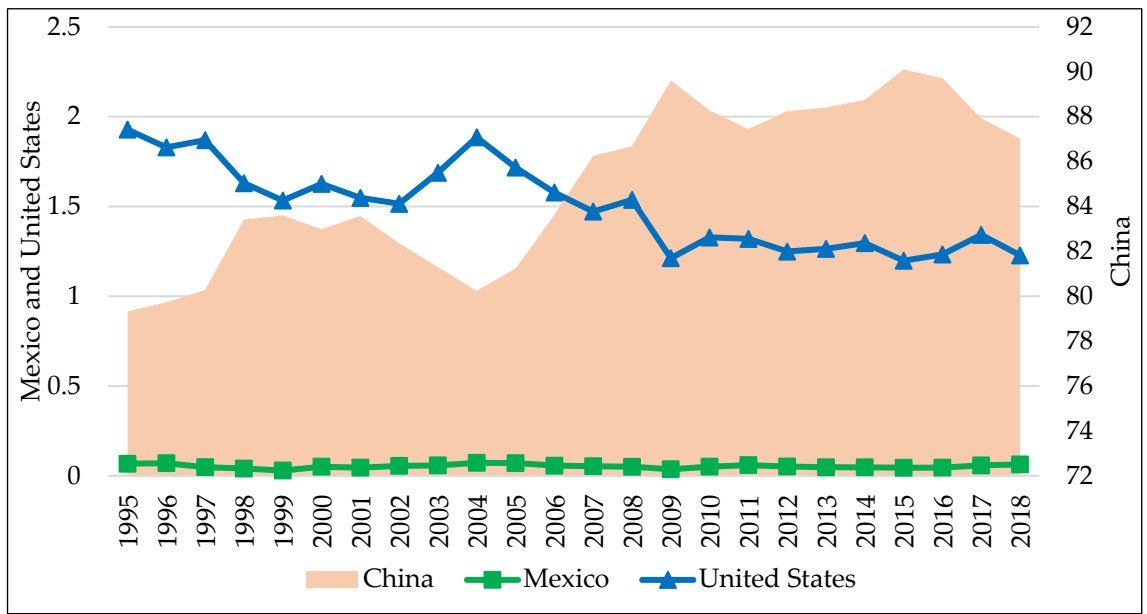

**Figure 5.** Origin of value added in China's textile and apparel gross exports (%). 1995–2018. Source: Authors based on TiVA (OECD 2021).

This behavior contrasts with the level and evolution of the domestic (Chinese) value-added share of Chinese textile exports, as it has a steady growth from 1995 to 1999 (79.3–83.6%) and increased just over four percentage points from 2000 to 2018 (82.9–87.0%).

Therefore, the United States has a descending trend in the origin of the added value of Chinese exports from 1995 to 2018. This predominance of China is not only due to the strategy of the complete package but also to the use of the strategy of devaluation of the currency against the dollar that has a positive effect on its textile exports whose purpose is to primarily affect the participation of the United States (Castro-Gonzáles and Mathews 2013).

However, there may be an increase in production costs in China, which would cause garment producers to have opportunities abroad to relocate their plants (Inomata and Taglioni 2019). Nevertheless, China has built a strong domestic market with a complete and independent manufacturing system, and through "Made in China" has been active in the global market internationalizing and inserting themselves into global supply chains (Cohen and Lee 2020; Ma et al. 2018).

### 3.3. The Contribution of Mexico, the United States, and China to the Final Global Demand for Textiles and Apparel

The previous section analyzed the evolution of the relative importance of Mexico, the United States, and China as the origin of the value added incorporated in their respective gross exports of textiles. However, for a better understanding of its implications, it is also interesting to study their importance in terms of their contribution to the final global demand for textile products.

This analysis shows the greater importance of China with a rising trend between 1995 and 2015 (from 6.2% to 38.3%), declining since then and contrasting notably with the declining share of the United States and Mexico, both with a share below 5% of global demand in 2018 (Figure 6).

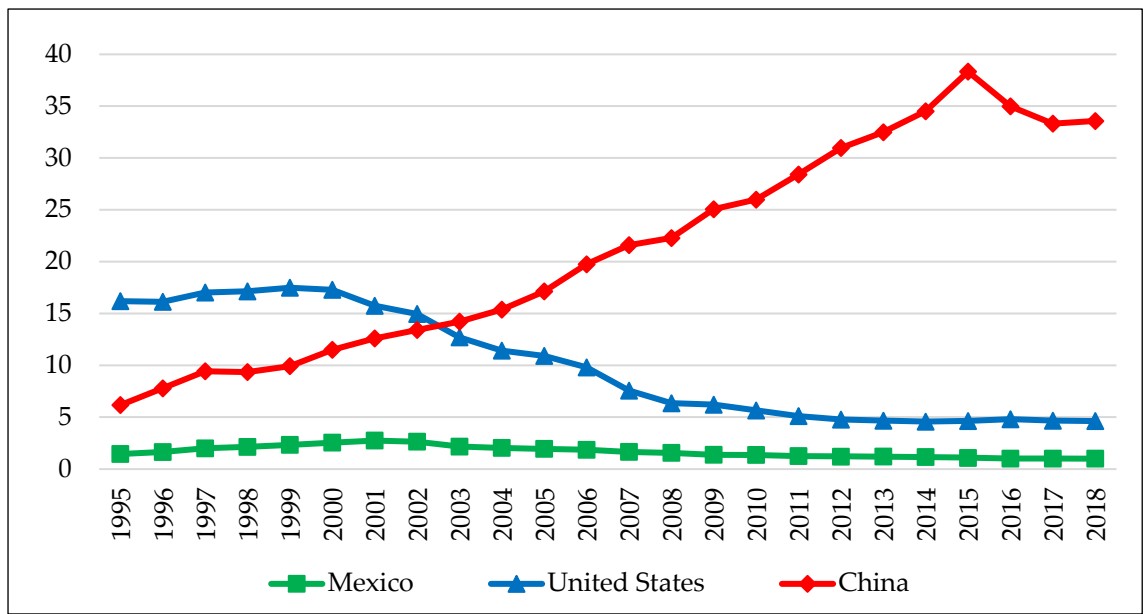

**Figure 6.** Participation of China, Mexico, and the United States as origin of the global final demand value added in the textile and apparel industry (%). 1995–2018 Source: Authors based on TiVA (OECD 2021).

Moreover, the comparison between the share in final global demand and the share in world exports shows opposite trends. The Mexican and the U.S. value-added loses relative importance, while value-added originating in China is becoming increasingly relevant, both at the export level and from the final global demand for textile products.

These results are in line with the general hypothesis that there is a process of consolidation of China's hegemony in the global textile market to the detriment of the participation of the United States and Mexico. Furthermore, these findings show that this process is being mainly conducted through the flows of value added in GVCs.

### 3.4. Econometric Results

This section presents the econometric estimations, first, for the U.S., Mexico, and China, and second, for the global case. These analyses aim to explore the influence of variables such as FDI, textile tariffs, and textile wages on GVC participation. Concerning participation, as indicated in the methodological section, a distinction is made between total and backward participation in GVCs.

As discussed in Section 2, two econometric models have been estimated to capture some of the determinants of total participation (TPART) and backward participation (BPART). The independent variables selected are the tariff rates applied to the textile sector, where TARIF1 refers to textile raw materials (model I) and TARIF2 refers to textile products (model II). The ratio of inward FDI stock to GDP (FDI) and labor costs to value

added in the textile industry (LABC). The main descriptive statistics for these variables are shown in Table 1.

**Table 1.** Main statistics of the variables included in the model.

| Variable | Mean | Median | S.D. | Min. | Max. |
|---|---|---|---|---|---|
| TPART | 50.5 | 50.4 | 9.76 | 25.9 | 81.9 |
| BPART | 28.0 | 27.4 | 10.1 | 3.91 | 54.8 |
| TARIF1 | 6.90 | 4.99 | 5.89 | 0.00 | 60.00 |
| TARIF2 | 14.86 | 11.65 | 10.80 | 0.00 | 100.00 |
| FDI | 70.6 | 32.2 | 198. | 0.613 | $1.99 \times 10^3$ |
| LABC | 54.1 | 55.6 | 15.2 | 19.8 | 133.5 |

Source: Author's elaboration based on information from OECD and UNCTAD.

The following subsections present the estimation results of the two models specified in Section 2. Section 3.4.1 presents the results for the group of countries formed by the United States, Mexico, and China, while Section 3.4.2 presents the results for the complete sample of 61 countries, adopting a global approach.

3.4.1. The Case of Mexico, the U.S., and China

Table 2 presents the estimation results for the specific case of Mexico, the U.S., and China. For each of the two models, there are two dependent variables: total (TTPART) and backward (BPART) GVC participation. The results for tariff variables are, apparently, counterintuitive, revealing a positive impact on GVC participation rates. However, this apparent contradiction may be justified by the heterogeneous nature and different profiles of international insertion of these three countries: Mexico, with higher backward participation in the textile and apparel GVC, and higher tariffs affecting textile manufacturing than the U.S. and China. Meanwhile, lower labor costs in Mexican textiles may be counterbalancing their higher tariffs. Regarding FDI, the results are in line with the positive effect expected for this variable, which reveals a positive and significant effect on backward participation.

**Table 2.** Results of the econometric estimation (time fixed effects) for total (TPART) and backward (BPART) GVC participation. Period: 1995–2018. Countries: 3 (China, Mexico, and United States) Number of observations: 72.

| | Total GVC Participation (TPART) | | Backward GVC Participation (BPART) | |
|---|---|---|---|---|
| | Model I | Model II | Model I | Model II |
| Constant | 36.9778 (3.3072) *** | 37.2433 (2.7351) *** | 10.6792 (3.4515) *** | 10.8526 (2.7861) *** |
| TARIF1 | 0.4301 (0.1175) *** | — | 0.4814 (0.1226) *** | — |
| TARIF2 | — | 0.2689 (0.0494) *** | — | 0.3058 (0.0503) *** |
| FDI | 0.0445 (0.0467) | 0.0569 (0.0422) | 0.1209 (0.0488) ** | 0.1360 (0.0430) *** |
| LABC | 0.0799 (0.0463) * | 0.0518 (0.0417) | 0.0636 (0.0483) | 0.0320 (0.0424) |
| $R^2$ | 0.83 | 0.86 | 0.95 | 0.96 |
| rho | 0.67 | 0.67 | 0.83 | 0.82 |

Source: Author's elaboration based on information from OECD and UNCTAD Note: The standard error is indicated in brackets. "***", "**" and "*" indicate significance at 0.01%, 0.05% and 0.1%, respectively.

3.4.2. The Global Case

This subsection extends the analysis carried out in the previous point to a broader set of countries. In particular, a total of 61 countries are considered, grouped into developed and developing countries, thus, capturing different participation patterns, according to

income level. Tables 3 and 4 present the main results of the estimation of the two models presented in expression (2) for three groups: full sample, developed economies, and developing economies (see Appendix B). It is of particular interest to consider both groups individually as the effects of FDI may vary on the performance of the textile industry, which is generally considered a central sector in the industrialization process of developing economies (Raei et al. 2019). As in Table 2, there are two cases for each model, considering total (Table 3) and backward (Table 4) GVC participation.

**Table 3.** Results of the econometric estimation (time fixed effects) for total GVC participation (TPART). Period: 1995–2018. Countries: 61. Number of observations: 1464.

| | Full Sample (61 Countries) | | Developed Economies (39 Countries) | | Developing Economies (22 Countries) | |
|---|---|---|---|---|---|---|
| | **Model I** | **Model II** | **Model I** | **Model II** | **Model I** | **Model II** |
| Constant | 51.0136 (0.8364) *** | 50.5985 (0.8683) *** | 50.5143 (1.2384) *** | 48.7253 (1.295) *** | 45.9248 (1.2792) *** | 44.0946 (1.3576) *** |
| TARIF1 | −0.1778 (0.0287) *** | — | 0.1352 (0.1024) | — | −0.1565 (0.0313) *** | — |
| TARIF2 | — | −0.0255 (0.0169) | — | 0.1828 (0.0421) *** | — | −0.0217 (0.0197) |
| FDI | −0.0023 (0.0008) *** | −0.0021 (0.0008) *** | −0.0022 (0.0008) | −0.0016 (0.0008) ** | 0.0438 (0.0126) *** | 0.0622 (0.0130) *** |
| LABC | 0.0166 (0.0152) | 0.0084 (0.0153) | 0.0461 (0.0202) ** | 0.0511 (0.0198) *** | −0.0168 (0.0231) | −0.0182 (0.0236) |
| $R^2$ | 0.83 | 0.83 | 0.80 | 0.81 | 0.79 | 0.78 |
| rho | 0.77 | 0.78 | 0.77 | 0.77 | 0.74 | 0.75 |

Source: Author's elaboration based on information from OECD and UNCTAD Note: The standard error is indicated in brackets. "***", and "*" indicate significance at 0.01%, and 0.05%, respectively.

**Table 4.** Results of the econometric estimation (time fixed effects) for backward GVC participation (BPART). Period: 1995–2018. Countries: 61. Number of observations: 1464.

| | Full Sample (61 Countries) | | Developed Economies (39 Countries) | | Developing Economies (22 Countries) | |
|---|---|---|---|---|---|---|
| | **Model I** | **Model II** | **Model I** | **Model II** | **Model I** | **Model II** |
| Constant | 30.6691 (0.8099) *** | 30.4297 (0.8515) *** | 31.1911 (1.1231) *** | 29.9897 (1.2062) *** | 26.7746 (1.3834) *** | 24.6514 (1.4766) *** |
| TARIF1 | −0.2621 (0.0278) *** | — | −0.5200 (0.0928) *** | — | −0.1909 (0.0339) *** | — |
| TARIF2 | — | −0.0653 (0.0166) *** | — | −0.0157 (0.0392) | — | −0.0299 (0.0214) |
| FDI | −0.0010 (0.0008) | −0.0009 (0.0008) | −0.0014 (0.0007) * | −0.0010 (0.0008) | 0.0481 (0.0136) *** | 0.0697 (0.0141) *** |
| LABC | −0.0151 (0.0147) | −0.0263 (0.0150) * | 0.0146 (0.0184) | −0.0023 (0.0184) | −0.0262 (0.0249) | −0.0281 (0.0257) |
| $R^2$ | 0.85 | 0.85 | 0.83 | 0.82 | 0.87 | 0.86 |
| rho | 0.81 | 0.82 | 0.79 | 0.79 | 0.84 | 0.84 |

Source: Author's elaboration based on information from OECD and UNCTAD Note: The standard error is indicated in brackets. "***" and "*" indicate significance at 0.01% and 0.1%, respectively.

According to the results presented in Tables 3 and 4, the tariff protection level is revealed as a key variable in the participation degree in GVCs, regardless of whether total or backward participation is considered a dependent variable. In the full sample models, for developing and developed countries in the case of backward participation, a higher level of tariff protection would lead to lower GVC participation in textiles through a trade-reducing effect concerning the economic costs associated with export and import flows.

This result is particularly relevant considering that the textile industry, especially in developing countries, is sensitive to internal and external cost changes. However, in the case of the developed countries presented in Table 3, the results suggest that higher tariff protection on textile products (but not on raw materials) would increase total GVC participation. These results may make sense insofar as the participation of developed countries in the textile value chain is more linked to design and branding tasks and less to the manufacture of textile products. Cost sensitivity does not only stand out among developing countries. As Lawless and Morgenroth (2019) point out, Brexit is an excellent example of how a change in trade tariffs can affect cost-sensitive sectors, such as food and textiles.

Something similar happens with the other variables incorporated in the model. In the case of FDI, it is generally significant in all cases, while labor costs are significant in the case of developing countries. Regarding FDI, a greater inflow would imply higher participation in GVC in the case of developing countries; however, in developed economies, a greater inflow of FDI would have the opposite effect. In this regard, it should be noted that developed countries are, in many cases, the final markets for textile products. In this sense, an increase in FDI in these countries could well reduce the global textile insertion of developed economies and strengthen their domestic market. Finally, it should be noted that when analyzing the influence of FDI on the total GVC participation without discriminating by sector of activity, the literature supports a significant and positive relationship (Martínez-Galán and Fontoura 2019; Okah Efogo et al. 2022).

Labor costs are significant, especially in developing countries, because their insertion in the textile sector occurs mainly through a cost strategy. Thus, a higher labor cost would imply lower participation in GVCs (Fukase 2013; Javed and Atif 2021; McCaig 2011). However, regarding developed countries, labor costs are not a useful variable to explain the insertion of the textile sector in GVCs since, in this group of countries, the industry does not depend on its competitiveness in terms of costs, but rather on its innovative capacity, design, and brand prestige (Padilha and Gomes 2016; Vila and Kuster 2007).

## 4. Conclusions

This paper is based on the recognition of the important role played by GVC in production processes, because of international fragmentation of production, which particularly affects manufacturing sectors such as the textile and apparel industry. In this sense, different conclusions can be drawn.

From a comparative perspective, the analysis of the GVC participation shows interesting results. China, and especially, Mexico and the United States present high levels of total participation, with different trends. However, from this sectoral approach, it also can be observed an accentuation of general GVC insertion patterns. In this sense, the United States and China show clearer forward participation, while Mexico follows a different pattern, with a predominance of backward participation. This feature is common in countries specializing in manufacturing and assembly tasks.

Another relevant finding is China's growing importance in the origin of the value-added incorporated in Mexican and U.S. textile exports. This result shows a steady increase in the backward linkages of Factory North America with the Asian giant. The contrary occurs in Mexico and especially the United States, which currently have residual importance as the origin of the value-added of China's textile exports and have even seen this importance decrease in the period of analysis (1995–2018).

These results are consistent with the characterization of the decline of Mexican and U.S. textile manufacturing, due to the presence of fragmented global trade flows not positively exploited. However, this contrasts with the innovative and competitive Asian textile and apparel industry, which adopts the full package strategy and FDI in a positive sense to increase its value-added with empowerment in productive fragmentation.

The analysis of the participation in the final global demand for textile products provides additional findings. The most remarkable result is that the Chinese share shows a clear and steady increase, approaching 40% of global demand. However, Mexico and the

United States show the opposite trend, becoming nearly residual. This contrasts with their respective shares in the textile GVC, where there are no significant differences in the levels of participation.

From an extended (61 countries) and explanatory perspective, the econometric analysis reveals tariff rates as a key factor for textile GVC participation, particularly in developing economies. However, the effect of other variables, such as FDI stock and labor costs, depends, to a large extent, on the group of countries considered. While among developed economies, FDI does not help to provide a simple understanding of their insertion in textile GVCs which, in developing economies, is a significant factor. This may be due to the type of tasks performed by each country. While the first group focuses on tasks such as branding and design, the developing countries mainly perform manufacturing tasks.

Although labor costs only reveal a significant and negative effect on backward participation for the full sample, this variable shows a negative impact mainly for developing countries, which is consistent with the expected influence of this factor. The counterintuitive result obtained for this variable for developed economies can be explained because the total GVC participation of these countries is driven by high labor costs (e.g., offshoring of manufacturing tasks to other countries), which causes a positive (and significant) effect on their total GVC participation. In contrast, the lower level of labor costs positively influences the GVC participation of developing economies.

Finally, this research highlights the profound changes the textile sector has undergone in a highly competitive and globalized context. In particular, it allows us to understand the changing role of economies in the international fragmentation of production, where countries follow different insertion patterns, with uneven results.

**Author Contributions:** Conceptualization, Ó.R.-M., H.C.-R. and A.L.G.P.; methodology, H.C.-R. and Ó.R.-M.; formal analysis, A.L.G.P. and H.C.-R.; investigation, A.L.G.P. and Ó.R.-M.; data curation, Ó.R.-M., A.L.G.P. and H.C.-R.; writing—original draft preparation, Ó.R.-M., A.L.G.P. and H.C.-R.; writing—review and editing, Ó.R.-M., A.L.G.P. and H.C.-R.; visualization, H.C.-R.; project administration, Ó.R.-M. All authors have read and agreed to the published version of the manuscript.

**Funding:** This research has been supported by the ICEDE research group, to which the authors belong, Galician Competitive Research Group financed by Xunta de Galicia (Ref. ED431C 2022/15).

**Data Availability Statement:** All data have been obtained from publicly available databases, such as TiVA (OCDE), UNCTAD, and WTO.

**Conflicts of Interest:** The authors declare no conflict of interest.

## Appendix A

*GVC Participation and Corresponding OECD TiVA Indicators (Extracted from Martins Guilhoto et al. 2022)*

Backward participation in GVCs, percentage (DEXFVApSH): Foreign VA embodied in exports, as % of total gross exports of the exporting country. This indicator is calculated for the total value of source and exporting industries; it is estimated as the ratio between the VA contents of imports from the source country $p$ and the gross exports of the exporting country $c$. This indicator is estimated as: $DEXFVApSH_{c,p} = EXGR\_BSCI_{c,p}/EXGR_c \times 100$, where $EXGR\_BSCI_{c,p}$ is the total VA from country $p$ embodied in the total exports of exporting country $c$, and $EXGR_c$ is the total gross exports of exporting country $c$.

Forward participation in GVCs, percentage (FEXDVApSH): Domestic VA embodied in foreign exports as a share (%) of total gross exports of the value-added source country. This indicator is calculated for the total value of source and exporting industries; it is estimated as the VA contents of exports originating in the source country, and embodied in the exports of the exporting country, divided by the gross exports of the source country. This indicator is estimated as follows: $FEXDVApSH_{c,p} = EXGR\_BSCI_{c,p}/EXGR_c \times 100$, where $EXGR\_BSCI_{c,p}$ is the total VA from country c embodied in the exports of country $p$, and $EXGR_c$ is the total gross exports of the value added source country $c$.

## Appendix B

**Table A1.** List of the 61 countries/territories and considered in the econometric analysis.

| Developed Economies (39) | | Developing Economies (22) | |
|---|---|---|---|
| Australia | Japan | Argentina | Thailand |
| Austria | Korea | Brazil | Tunisia |
| Belgium | Latvia | Bulgaria | Turkey |
| Canada | Lithuania | Cambodia | |
| Taiwan | Malta | Chile | |
| Croatia | The Netherlands | China | |
| Cyprus | New Zealand | Colombia | |
| Czech Republic | Norway | Costa Rica | |
| Denmark | Poland | India | |
| Estonia | Portugal | Kazakhstan | |
| Finland | Saudi Arabia | Laos | |
| France | Singapore | Malaysia | |
| Germany | Slovak Republic | Mexico | |
| Greece | Slovenia | Morocco | |
| Hong Kong | Spain | Peru | |
| Hungary | Sweden | Philippines | |
| Iceland | Switzerland | Romania | |
| Ireland | United Kingdom | Russian Federation | |
| Israel | United States | South Africa | |
| Italy | | | |

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
