# Peer review of "The Global Textile and Apparel Value Chain: From Mexico–US–China Linkages to a Global Approach"

_economies, doi:10.3390/economies10100258_

Round 1
Reviewer 1 Report
A well-designed macroeconomic model of GVC participation in the textile sector of the US-Mexico-China trifecta based on longitudinal TiVA data is a worthwhile endeavour.
Unfortunately, this study does not achieve this. The overly broad title demonstrates its fundamental weakness which is a lack of clear scientific purpose. I had to re-read the introduction several times in search of a precise formulation of the research question. A statement finally appeared on page 9/line 373, to the effect: "the main research question underlying this paper: Is China consolidating its leadership as the main supplier of value added to the global textile and apparel value chain developing an increasingly important role as a value creator in this global industry?" If this is the main research question, it should appear in the first section shortly after a summary of the study's motivation. The formulation as it stands is vague and rambling.
Perhaps most seriously: this research question does not clearly justify the econometric model which was computed based on the selected TiVa data. Please make clear whether the textile GVCs in question involve the trade flows between the firms which operate across the borders of US-China-Mexico or some combinatoric of US-China vs. US-Mexico vs. China-Mexico (all of which exist and are distinct in their design strategies). These distinctions are key to the research setup and interpretation of the data. It would help to provide some empirical examples here.
Most of the references in the paper are relevant but they are far from comprehensive. The study has not been properly positioned in the stream of extant research. The literature is presented in a superficial and sloppy way, with the basic terminology defined in confusing prose. At the very least, the paper must explain the logical emergence of GVCs more precisely than "The large increase in the demand for textile products has opened the door to complex production processes that met international quality standards and paved the way to produce complete packages, stimulating the formation of linkages between members of the production chain and providing opportunities for domestic producers. This has led to the emergence of Global Value Chains (GVCs), a complex phenomenon reflecting the im portance of global production linkages for access to new technologies, training and innovation" (lines 27-32). This is simply not a correct explanation of the forces which led to the emergence of GVCs.
The author should refer to the seminal definition by Gereffi & Fernandez-Stark 2011 who specify the 4 key characteristics of an input-output structure, geographical context, a governance structure and an institutional context. There should be a brief summary of why this characterization is new and noteworthy compared to orthodox definitions of international trade. Baldwin provides a detailed narrative explanation of the forces of information and logistics technology, political liberalization, and modularization which led to the emergence of GVCs (Baldwin, R. (2016). The Great Convergence: Information Technology and the New Globalization. Cambridge: Belknap Press).
There are other fundamental studies which should be referenced, which have discussed the consequences of the dis-integration of the production function on conventional trade policy instruments (like tariffs). The most important consequence of de-verticalization is the explosion in the trade volume of intermediates and not finished goods between countries. Since intermediates in a GVC can cross borders several times, trade protection measures like tariffs magnify costs exponentially (for details see OECD (2013). Interconnected Economies; Benefiting from Global Value Chains. Available at: http://dx.doi.org/10.1787/9789264189560-en). There are surely more recent studies of trade cost amplification but this old one has seminal quality. Reversing this logic reveals the power of integration, since vertical specialization can magnify tariff reductions into large increases in trade (see Yi, Kei-Mu. (2003). Can vertical specialization explain the growth of world trade? Journal of political Economy 111.1, 52-102.) Before any econometric model is set up to describe the effect of Tariffs on GVC participation, these basic principles must be presented as a point of departure. For the paper to make any contribution, its interpretation of the model must return to them and extend them.
The author should also revisit and revise the following incorrect statements in the text:
Lines 51-52: "In addition, the fact that the textile industry does not undergo frequent or radical changes in production methods allows suppliers to plan production in advance." The production methods of textiles do not allow suppliers to plan their production more in advance than other industries. This vast generalization should be broken down into verifiable facts, ideally with some empirical illustration.
Lines 146-147: "Therefore, achieving efficiency in GVC depends largely on how firms interconnect, either through vertical integration with the aim of reducing operating costs and better coordinating the supply chain (WTO, 2019) or by generating efficient and robust GVCs." There are several illogical, nested statements here.GVCs are actually the outcome of vertical dis-integration. Achieving efficiency in a GVC is achieved by generating efficient and robust GVCs????
It is not meaningful to review the econometric model in more detail before these fundamentals have been established. Like the preceding sections, the model and its interpretation can be made more clear.
The author may want to consider breaking the paper into two sections. The first would consider the 3-country model and the second analyze a more global system. At the moment the logical transition from the first to the secton dataset is not made clear.
Author Response
The authors would like to thank to the reviewer very much for his/her valuable comments made to the previous version of the paper. Thanks to these comments and suggestions, the revision work addressed both theoretical and methodological issues that brought clearness and robustness to the analysis. In particular, the inclusion of a new approach based on econometrics analysis has been fundamental and brought robustness to the study.
According to the valuable suggestions from the reviewer, this new revised version includes the following major changes:
- The aim of the research has been clarified both in the abstract and in the introduction.
- The paper has been reorganized. The introduction and the literature review have been synthesized into a single section, making the text more coherent.
- The recommended references on the role of tariffs in the evolution of trade flows have been added.
- The econometric analysis has been extended and the section has been divided in two parts. A first one dedicated to the case of Mexico, US and China and a second one dedicated to the general case with more countries (61). Two new tariff variables concerning textile raw material and textile finished products have also been included.
- All minor changes suggested by the reviewer have been addressed.
In addition, a careful revision of the English has been made with the assistance of a native speaker.
The authors hope that these changes result satisfactory and that the paper meets the criteria for publication.
Kind regards

Reviewer 2 Report
This paper looks at the participation of China, Mexico and the United States in the textiles and apparel global supply chain using backward and forward participation indicators from TiVA for the years 1995 to 2018. The stylized facts from this section are interesting. The authors show that China has increased its share of value added in Mexican and US textile and apparel exports, as well as increasing the domestic share of value added in Chinese exports. Mexico’s value added in US and Chinese textile exports is stagnant throughout the period, while domestic value added in textile and apparel exports in Mexico drops in the middle of the period but recovers to 1995 levels. China’s share of global final demand in textiles and apparel increased significantly, while the US share experiences a moderate decline and Mexico’s share experiences a small decline. The authors then undertake an econometric analysis of what drives total and backward participation in the textile and apparel industry for 61 countries using the TiVA indicators and splitting the sample into two country groups: developed and developing. The authors test the impact of tariffs, inward FDI and labor costs on GVC participation in the full sample and the two country groups. Tariffs have the expected negative effect. Inward FDI has a negative and significant effect in developed countries and a positive and significant effect in developing countries. Labor costs have the expected negative effect when significant. Overall, the paper presents some new facts using the TiVA data and an interesting econometric exercise on what is driving GVC participation globally.
General comments:
The paper needs to be better organized and a more defined link between the descriptive statistics and the econometrics needs to be made.
Organization: The paper would read better if the descriptive statistics using the TiVA data were introduced in section 2, then the data and methodology, then the econometric results.
The introduction and current section 2 on GVCs can be condensed considerably into the main points that are relevant to the paper. Likewise, the presentation of the descriptive statistics using the TiVA data can also be condensed into the main points without providing as much background information and references.
Link between descriptive statistics and econometrics: Currently, there is only a loose link between the descriptive statistics presented using the TiVA data, which are only for the US, China and Mexico, and the econometrics using the full set of countries. The paper would be more cohesive if the descriptive statistics presented were for the full set of countries and did not focus on specific countries OR if the econometrics was specific to the case of the US, China, Mexico + textiles and apparel industry. Because the descriptive statistics are not linked to the econometrics, it is not clear what the main research question is in the paper.
The econometric analysis could be extended and improved. Are any control variables used? How robust are these results to other explanations or robustness checks? It is not clear whether country fixed effects are included in the estimation. If country level fixed effects are included in the estimation, then the coefficients are estimated only from variation over time. It would be good to see the results without any fixed effects as the first columns in the table to compare. What do these results mean for the case of China, the United States and Mexico?
Specific Comments:
1. Line 162: should be developing and not developed
2. Why use manufactures tariff when all variables refer to the textile sector? It is possible to calculate an average tariff for just the textiles and apparels sector, which would be appropriate for this analysis because imported inputs like yarn and exported final products like clothing all fall under the textiles and apparel sector.
3. Line 256: Consider using “country” instead of “individual” to reinforce that it is a country level effect.
4. Lines 280 and 281: Original Equipment Manufacturing is listed twice
5. Lines 315-318: Is there anything about labor markets, wages, unions, etc in Mexico that might also account for Mexico’s position in GVCs?
6. Lines 373 to 376: Another way to state this question: Is a rise in China’s involvement in the textiles and apparel GVC responsible for the decline in US and other countries’ value added in Mexico’s textile exports? This is an interesting research question to answer econometrically, but the paper only presents econometrics at a more general level. Likewise, another interesting and related research question: is the rise of China in the global textile and apparel GVC responsible for the decline in participation in this GVC from other countries?
7. Explain more what the expected sign is for each group of countries for all independent variables.
Author Response
The authors would like to thank to the reviewer very much for his/her valuable comments made to the previous version of the paper. Thanks to these comments and suggestions, the revision work addressed both theoretical and methodological issues that brought clearness and robustness to the analysis. In particular, the inclusion of a new approach based on econometrics analysis has been fundamental and brought robustness to the study.
According to the valuable suggestions from the reviewer, this new revised version includes the following major changes:
- The paper has been reorganized. The introduction and the literature review have been synthesized into a single section, making the text more coherent.
- Some references of minor relevance to the aim of the research have been removed.
- The econometric analysis has been extended and the section has been divided in two parts. A first one dedicated to the case of Mexico, US and China and a second one dedicated to the general case with more countries (61). Two new tariff variables concerning textile raw material and textile finished products have also been included.
- All minor changes suggested by the reviewer have been addressed.
- A specific comment about labor market and labor costs have been included in the text to clarify the Mexico’s participation in GVCs.
- The authors are grateful for the comment and suggestion from the reviewer about the Chinese role in the textiles and apparel GVC and its possible effects on the value-added origin of Mexico’s textile exports. However, we believe that this interesting research question would constitute a new research and would require a deeper econometric analysis that exceeds the scope of this paper.
In addition, a careful revision of the English has been made with the assistance of a native speaker.
The authors hope that these changes result satisfactory and that the paper meets the criteria for publication.
Kind regards

Round 2
Reviewer 1 Report
Dear authors,
Thank you for your thoughtful consideration of the first review. The paper has improved a great deal and now presents both a coherent argument and more clarity of method. The references provide a good summary of the extant literature. The written style is also smoother and easier to follow. Please do a last check of the prose since there remain a few baffling sentences like the one in 85-88: "In addition to the fact that fragmentation of production is regardless of the form GVCs take implies a crossing of borders resulting in a greater international division of labor and greater benefits from specialization that the textile industry takes advantage of". Otherwise, the work is ready to publish.
Author Response
Dear Reviewer,
Once again, thank you for your comments. Thanks to them, the quality of our paper has greatly improved. We have made a complete revision of the writing, improving the use of English and correcting several typos.
We hope you will be able to see our work published soon.
Best regards,
The Authors
Reviewer 2 Report
The reorganized paper reads much better. The inclusion of the different tariffs make sense in a GVC analysis. A few clarifications/edits are needed:
1. Do the specifications have country or time fixed effects (or both)?
2. Line 561 - there is a period in the middle of a sentence
3. Overall editing: the writing could be more cohesive and concise.
Author Response
Dear Reviewer,
The authors would like to thank the Reviewer for the valuable comments on the previous versions of the paper. Thanks to them, the quality of our manuscript has greatly improved. In addition, we have addressed the suggested minor changes, specifying the adoption of time fixed effects and performing a complete revision of the English and wording, improving them and correcting several typos. Thus, we expect the paper to meet the requirements for publication.
Best regards,
The Authors